# Immunogenic Responses Elicited by a Pool of Recombinant *Lactiplantibacillus plantarum* NC8 Strains Surface-Displaying Diverse African Swine Fever Antigens Administered via Different Immunization Routes in a Mouse Model

**DOI:** 10.3390/vaccines13090897

**Published:** 2025-08-25

**Authors:** Assad Moon, Hongxia Wu, Tao Wang, Lian-Feng Li, Yongfeng Li, Zhiqiang Xu, Jia Li, Yanjin Wang, Jingshan Huang, Tianqi Gao, Yuan Sun, Hua-Ji Qiu

**Affiliations:** State Key Laboratory for Animal Disease Control and Prevention, Harbin Veterinary Research Institute, Chinese Academy of Agricultural Sciences, Harbin 150069, China; 2021y90100029@caas.cn (A.M.); wuhongxia@caas.cn (H.W.); wangtao07@caas.cn (T.W.); lilianfeng@caas.cn (L.-F.L.); liyongfeng@caas.cn (Y.L.); zhiqiangxu01@163.com (Z.X.); lijia0007070@163.com (J.L.); wangyanjin1996@126.com (Y.W.); hjs45333@163.com (J.H.); 19990736668@163.com (T.G.)

**Keywords:** African swine fever, lactic acid bacteria, *Lactiplantibacillus plantarum*, surface display, immunological evaluation

## Abstract

**Background:** African swine fever (ASF) is a highly contagious and often deadly disease that poses a major threat to swine production worldwide. The lack of a commercially available vaccine underscores the critical need for innovative immunization strategies to combat ASF. **Methods:** Six ASFV antigenic proteins (K78R, A104R, E120R, E183L, D117L, and H171R) were fused with the *Lactiplantibacillus plantarum* WCFS1 surface anchor LP3065 (LPxTG motif) to generate recombinant *Lactiplantibacillus plantarum* NC8 (rNC8) strains. The surface expression was confirmed using immunofluorescence and Western blotting assays. Additionally, the dendritic cell-targeting peptides (DCpep) were co-expressed with each antigen protein. Mice were immunized at a dosage of 10^9^ colony-forming units (CFU) per strain per mouse via intragastric (I.G.), intranasal (I.N.), and intravenous (I.V.) routes. The bacterial mixture was heat-inactivated by boiling for 15 min to destroy viable cells while preserving antigenic structures. I.V. administration caused no hypersensitivity, confirming the method’s safety and effectiveness. **Results:** Following I.G. administration, rNC8-E120R, rNC8-E183L, rNC8-K78R, and rNC8-A104R induced significant levels of secretory immunoglobulin A (sIgA) in fecal samples, whereas rNC8-H171R and rNC8-D117L failed to induce a comparable response. Meanwhile, rNC8-D117L, rNC8-K78R, and rNC8-A104R also elicited significant levels of sIgA in bronchoalveolar lavage fluid (BALF). Following I.N. immunization, rNC8-E120R, rNC8-K78R, and rNC8-A104R significantly increased sIgA levels in both fecal and BALF immunization. In contrast, I.V. immunization with heat-inactivated rNC8-K78R and rNC8-A104R induced robust serum IgG titers, whereas the remaining antigens elicited minimal or insignificant responses. Flow cytometry analysis revealed expanded CD3^+^CD4^+^ T cells in mice immunized via the I.N. and I.G. and CD3^+^CD4^+^ T cells only in those immunized via the I.N. route. Th1 responses were also significant in the sera of mice immunized via the I.G. and I.N. routes. **Conclusions:** The rNC8 multiple-antigen cocktail elicited strong systemic and mucosal immune responses, providing a solid foundation for the development of a probiotic-based vaccine against ASF.

## 1. Introduction

African swine fever (ASF), caused by the African swine fever virus (ASFV), poses a major threat to global swine health and is classified as a notifiable disease by the World Organization for Animal Health (WOAH) [1,2]. African swine fever virus (ASFV), a member of the *Asfivirus* genus within the *Asfarviridae* family, primarily replicates in the cytoplasm of infected macrophages, although an initial replication phase occurs within the nucleus [3]. The ASFV genome spans approximately 170 to 194 kilobase pairs (kb), encoding between 150 and 167 open reading frames (ORF) with a conserved central region of approximately 125 kb, flanked by variable terminal regions [4]. Two-dimensional electrophoresis identified a minimum of 28 structural intracellular particles and 54 purified extracellular proteins [5]. Over 100 viral proteins have been identified in infected porcine macrophages [6]. Previous studies have shed light on vectored and subunit vaccines against ASF, emphasizing their safety advantages over other vaccine development technologies. Many structural proteins, such as p30, p54, p72, EP153R, D117L, and CD2v, have been used to develop vaccines against ASF [7]. Nonetheless, the utilization of various subunit vaccine strategies, specifically those employing p30, CD2v, and p72 as antigens, has exhibited restricted advancement and yielded inconsistent results [8]. Moreover, relying solely on the antigen has not proven to be a promising strategy to protect animals against virus challenge [9]. The previously inactivated vaccines showed limited efficacy against ASF. Efforts are being made to develop more effective live attenuated vaccines, i.e., gene-deleted, subunit, DNA, and vector-type vaccines. Vietnam’s ASFV-G-ΔI177L is the first commercial ASF vaccine developed in that country, but strain attenuation does not guarantee virulence due to extensive genotype-specific defense mechanisms. The viral genome encodes the synthesis of numerous proteins essential not only for viral assembly but also for critical functions such as DNA replication, repair mechanisms, and the intricate control of genetic expression [10]. Moreover, the ASFV genetic blueprints contain a range of proteins cleverly engineered to thwart host immune responses, effectively disrupting type I interferon signaling and manipulating cellular death pathways in its favor [11].

A systematic evaluation of approximately 30% of ASFV antigens was conducted using a multi-stage immunization strategy, combining DNA priming with recombinant vaccinia virus boosting [12]. This comprehensive methodology aimed to systematically assess each antigenic component’s immunological potential in pigs, advancing our comprehension of ASFV immunobiology and aiding in vaccine target selection. The viral D117L and E120R proteins were identified as promising candidates for eliciting protective antibody responses. Furthermore, viral proteins p54 (*E183L*) and CD2v (*EP402R*) effectively stimulated specific lymphocytes, eliciting potent cellular immunity [11,13].

The NC8 strain of *Lactobacillus* has emerged as a promising mucosal vaccine vector. This is because it has a Generally Recognized as Safe (GRAS) status and demonstrated immunomodulatory properties [14]. For instance, NC8, engineered to express the porcine epidemic diarrhea antigenic proteins and the coxsackievirus B4 protein, effectively induced the production of mucosal antibodies and a potent cell-mediated immune response [15,16]. Recently, a study evaluated the immune responses induced by a recombinant NC8 strain expressing the ASFV p54 (*E183L*) protein fused with the porcine IL-21 in mice [17].

In this study, a mixture of six recombinant NC8 bacterial strains, each expressing one of the antigenic proteins of ASFV (*E120R*, *E183L*, *H171R*, *D117L*, *K78R*, and *A104R*), was selected and mixed together to formulate an ASF vaccine candidate. We then verified the immune responses of these candidates in mice via the I.V., I.G., or I.N. routes. The goal of this study is to provide a basis for the development of a vaccine using LAB as a carrier for ASFV antigens by optimizing the route of administration. By exploring these vaccine candidates, we aim to present new avenues for advancing the development of an effective vaccine candidate against ASF. The overall objective is to contribute to the control of ASF.

## 2. Materials and Methods

### 2.1. Bacterial Strains, Viruses, and Plasmids

The host strain *Lactococcus lactis* subsp. *cremoris* NC8, the pMG36e shuttle plasmid, and the ASFV HJL/2018 strain (GenBank accession number: MK333180.1) were preserved in our laboratory.

### 2.2. Construction of the Recombinant Plasmids Surface-Expressing the ASFV Antigens

The LP3065 peptide of *L. plantarum* WCFS1 (GenBank accession number: AL935263.2) and the pMG36e-LP3065 vector were reported in our previous study [18]. Moreover, 6× His-tag and Strep-tag were fused to each target protein, and the genes of 6× His-tag and Strep-tag have already been inserted into the pMG36e-LP3065 vector. To enhance the expression of viral proteins of ASFV in the NC8 host strain, all the corresponding viral genes were codon-optimized and synthesized by GenScript Company. Following this, the six ASFV antigen genes (*E120R*, *E183L*, *D117L*, *H171R*, *K78R*, and *A104R*) were amplified using specific primers shown in Table 1. All six ASFV antigen gene segments were ligated into the pMG36e-LP3065 vector between the *Xho*I and *Bam*HI restriction sites by T4 DNA ligase at 16 °C to obtain the recombinant plasmids pMG36e-LP3065-E120R, pMG36e-LP3065-E183L, pMG36e-LP3065-D117L, pMG36e-LP3065-H171R, pMG36e-LP3065-K78R, and pMG36e-LP3065-A104R.

### 2.3. Construction of the Recombinant NC8 (rNC8) Strains

Approximately 10 μL of each recombinant plasmid was mixed with 50 μL of electro-competent *Lactococcus lactis* subsp. *cremoris* NC8 on ice for 2 min, and then transferred to a pre-chilled 0.2 cm electroporation cuvette. Electroporation was performed using settings of 2000 V, 25 μF, and 200 Ω, resulting in a pulse of 4 to 5 milliseconds. After electroporation, the cells were resuspended in 700 μL of ice-cold, antibiotic-free MRS broth and incubated on ice for an additional 5 min. Following a 1-h incubation at 37 °C, cultures were centrifuged at 6000× *g* for 2 min. The resulting cell pellets were resuspended in 200 μL of MRS medium and plated on MRS agar containing 10 μg/mL erythromycin. After incubation at 37 °C, single colonies were picked and cultured in MRS broth. Plasmids were extracted and sequenced to confirm the insertion of the target genes. The recombinant strains with confirmed sequences designated as rNC8-E120R/E183L/D117L/H171R/K78R and rNC8-A104R were stored at −80 °C for further experimentation.

### 2.4. Immunofluorescence Identification of Recombinant Proteins on the Surface of rNC8 Strains

For immunofluorescence analysis, rNC8-E120R/E183L/D117L/H171R/K78R, rNC8-A104R, and an empty vector control (rNC8-pMG36e) were grown overnight in MRS broth containing 10 μg/mL erythromycin. The grown bacterial pellets were collected by centrifugation, washed twice with phosphate-buffered saline (PBS), and fixed with 4% paraformaldehyde for 30 min at room temperature (RT). Anti-His-tag antibody (catalog no. K200060M, Solarbio, Beijing, China) and Alexa Fluor 488 goat anti-mouse IgG (catalog no. A-11001; ThermoFisher Scientific, Agawam, MA, USA) were used as the primary and secondary antibodies, respectively. The bacterial pellets bearing each plasmid were incubated with the respective antibodies at 37 °C for 1 h and 45 min. Following incubation, samples were washed three times with PBS. Subsequently, 2 µL of each bacterial suspension was placed onto a glass slide, air-dried, and visualized under a fluorescence microscope.

### 2.5. Western Blotting Analysis of Recombinant Proteins Expressed on the Surface of rNC8 Strains

The recombinant strains, each carrying the respective plasmids, were cultured anaerobically overnight at 37 °C in 10 mL of MRS medium supplemented with 10 μg/mL erythromycin. The overnight cultures were centrifuged at 5000× *g* for 10 min at 4 °C. The resulting pellets were washed three times with cold PBS and lysed in 100 μL of radioimmunoprecipitation assay (RIPA) buffer at 4 °C for 30 min and centrifuged (10,000× *g*, 10 min). Proteins in the supernatants were separated by SDS-PAGE and transferred onto a PVDF membrane, which was blocked with 1% skim milk for 3 h at room temperature. The membrane was then incubated with anti-His-tag antibodies diluted 1:2000 for 2 h at RT. The PVDF membrane was washed twice and subsequently incubated with IRDye 680RD goat anti-mouse IgG secondary antibody (catalog no. C50113-06, Li-Cor, Lincoln, RI, USA) at a 1:10,000 dilution for 45 min at RT. The antibody-treated membrane was rinsed twice with PBS, and protein detection was performed using a Li-Cor Odyssey dual-color infrared imaging system.

### 2.6. Immunization of Mice

For mouse immunization, each rNC8 strain was anaerobically cultured overnight at 37 °C in MRS broth supplemented with 10 μg/mL erythromycin. The next day, bacterial cells were harvested by centrifugation (5000× *g*, 10 min, 4 °C), washed twice with sterile PBS (pH 7.4), and resuspended in PBS to an OD_600nm_ of 1.0. The strains were then pooled to prepare the rNC8-LP3065-ASFV mix.

For the immunization, specific pathogen-free (SPF) female BALB/c mice were employed. All mice were housed under pathogen-free conditions at an ambient temperature of approximately 25 °C, with *ad libitum* access to water and food throughout the experimental period. Forty six-week-old mice were distributed into nine groups, i.e., rNC8-LP3065-ASFV-mix (I.V., I.G., and I.N.), rNC8-pMG36e (I.V., I.G., and I.N.), and PBS (I.V., I.G., and I.N.). In our study, to avoid hypersensitivity reactions from live bacteria, the bacterial mixture was heat-inactivated by boiling in distilled water for 15 min. This process ensured the destruction of viable cells while preserving key antigenic structures. After cooling, the inactivated preparation was administered via the I.V. route. No signs of hypersensitivity, such as local inflammation or systemic reactions, were observed during the immunization schedule, confirming the safety and effectiveness of the inactivation method. The immunization strategy involved administering rNC8-LP3065-ASFV-mix (10^9^ CFU/strain/mouse) via I.V., I.G. (200 µL), and I.N. (20 µL) routes to each group. For I.N. immunization, the mice were anesthetized with 2.5% Avertin (0.02 mL/g body weight).

### 2.7. Enzyme-Linked Immunosorbent Assay (ELISA)

IgG and IgA antibodies in the sera, intestines, and lungs of the immunized mice were examined by ELISA. The intestines and lungs collected from the immunized mice were weighed (approximately 0.1 g) and homogenized in 100 µL of DMEM by the Tissue Lyser II (Biorepublic, Beijing, China), and the supernatants from each tissue sample were collected for detection of IgG and IgA by ELISA. The purified ASFV antigenic proteins (*E120R*, *E183L*, *D117L*, *H171R*, *K78R*, and *A104R*) were coated onto 96-well cell culture plates to detect the antibodies in sera and supernatants from intestinal and lung samples of immunized mice [14]. Serum cytokines IL-2 (catalog no. SEA073Mu), IL-4 (catalog no. SEA077Mu), IL-10 (catalog no. SEA056Mu), and IFN-*γ* (catalog no. SEA049Mu) levels were analyzed using a commercially available ELISA kit (Cloud-Clone, Wuhan, China).

### 2.8. Isolation of T Lymphocytes from the Spleens of the Immunized Mice

The splenocytes were isolated from the spleens of immunized mice (*n* = 3) at designated time points. The spleens were dissected into small fragments, homogenized in tissue homogenization buffer, and gently passed through a cell strainer (70 µm) to obtain a single-cell suspension. The splenocytes collected from the filtrate were carefully layered onto lymphocyte separation medium and subjected to centrifugation at 500× *g* for 30 min. The resulting lymphocyte layer at the interface was harvested, washed twice with PBS, and treated with erythrocyte lysis buffer (catalog no. BL503B; Biosharp Life Sciences, Hefei, China). The harvested cells were then counted and prepared for subsequent experiments.

### 2.9. Proliferation Assay of Splenic T Lymphocytes

The splenocytes isolated from immunized mice were cultured in 96-well cell culture plates at a density of 10^6^ cells/mL (50 μL per well) and assessed for proliferation using a CCK-8 assay. The cells were stimulated with ASFV at 10^5^ TCID_50_ and with 10 μg/mL concanavalin A as a positive control or culture medium as a negative control. MTT reagent was added to each well at a final concentration of 0.5 mg/mL, followed by incubation at 37 °C in the dark for 3 h. Absorbance was measured at OD450_nm_ using a microplate spectrophotometer.

### 2.10. Flow Cytometry Analysis of T-Cell Surface Markers

Single-cell suspensions of splenocytes from experimental and control groups were prepared and adjusted to a concentration of 10^6^ cells/mL. The cells were stained with PE/Cy5-conjugated anti-mouse CD3 (catalog no. 100273; BioLegend, London, UK), APC/Cy7-conjugated anti-mouse CD8*α* (catalog no. 100713; BioLegend, London, UK), and FITC-conjugated rat anti-mouse CD4 (catalog no. 557307; BD Biosciences, Shanghai, China) on ice for 30 min. Subsequently, unbound antibodies were removed by washing the cells with PBS. The stained cells were detected using flow cytometry (ApogeeFlow System, Northwood, UK) and analyzed by using the Apogee histogram software 1.25.52.

### 2.11. Ethics Statements

All experimental procedures involving ASFV were performed in BSL-3 and others in BSL-2 biocontainment facilities at the Harbin Veterinary Research Institute (HVRI), Chinese Academy of Agricultural Sciences (CAAS), with approval from China’s Ministry of Agriculture and Rural Affairs. The study complied with the Animal Welfare Act and followed experimental animal care guidelines, as approved by HVRI’s Laboratory Animal Welfare Committee (Approval No. 230724-01-GR).

### 2.12. Statistical Analysis

All experiments were performed in triplicate, and data were presented as mean ± standard deviation (SD). Statistical analyses were performed using GraphPad Prism version 8.3 (GraphPad Software, San Diego, CA, USA). The statistical significance was defined as ns: not significant *p* ≥ 0.05, * *p* < 0.05, ** *p* < 0.01, *** *p* < 0.001.

## 3. Results

### 3.1. Surface-Expression Analysis for LP3065 Fused with the ASFV Antigens

The recombinant plasmids encoding ASFV antigens were constructed using the LP3065 surface anchoring polypeptide, as illustrated in the schematic diagram (Figure 1A). The recombinant plasmids were transformed into NC8 expression bacteria, yielding the recombinants rNC8-LP3065-E120R/E183L/D117L/H171R/K78R/A104R. Confocal laser scanning microscopy revealed green fluorescence on the bacterial cell surfaces (Figure 1B), confirming the ASFV antigen expression. No fluorescence was observed on the negative control recombinant rNC8-pMG36e bacteria.

Moreover, Western blotting confirmed the target proteins LP3065-E120R (33 kDa), LP3065-E183L (33 kDa), LP3065-D117L (32 kDa), LP3065-H171R (39 kDa), LP3065-K78R (28 kDa), and LP3065-A104R (31 kDa) are expressed at the anticipated molecular weight (Figure 1C and Appendix A).

### 3.2. Analysis of IgG Antibodies Production by rNC8-LP3065-ASFV-mix in Mice

To further evaluate the role of rNC8-LP3065-ASFV strains in humoral immune response, mice were immunized with heat-inactivated rNC8-LP3065-ASFV-mix via the I.G., I.N., and I.V. routes. Mice administered rNC8-pMG36e or PBS served as negative controls. Blood, fecal, and BALF samples were collected at designated time points (Figure 2).

The results indicated that oral immunization with rNC8-LP3065-ASFV-mix induced higher IgG levels in the BALF compared with the rNC8-pMG36e and PBS control groups from 6 dpi to 29 dpi. I.G. immunization significantly elevated IgG levels in the mice immunized with rNC8-LP3065-K78R and rNC8-LP3065-A104R, while rNC8-LP3065-H171R, rNC8-LP3065-E120R, and rNC8-LP3065-E183L induced lower antibody responses. In contrast, I.N. immunization generated higher levels of IgG responses to rNC8-LP3065-K78R, rNC8-LP3065-A104R, and rNC8-LP3065-E183L, but weaker responses to rNC8-LP3065-H171R and rNC8-LP3065-D117L. No significant IgG was observed for rNC8-LP3065-E120R (Figure 3). The levels of IgG antibodies in rNC8-LP3065-A104R and rNC8-LP3065-K78R were substantially significant, while insignificant levels of IgG were observed in the mice I.V. immunized with rNC8-LP3065-E183L, rNC8-LP3065-E120R, and rNC8-LP3065-H171R. These data indicated that immunization of rNC8-LP3065-ASFV-mix can induce robust humoral immune responses via I.V. groups and I.G. gavage but not via I.N. administration.

### 3.3. rNC8-LP3065-ASFV-mix Induced High-Level Secretory IgA Antibodies in Mice

We investigated whether rNC8-LP3065-ASFV-mix could enhance the sIgA antibody responses triggered by I.G. or I.N. immunization. BALB/c mice received three inoculations, each consisting of three daily doses, either with rNC8-LP3065-ASFV-mix or rNC8-pMG36e. Over time, we observed the development of anti-ASFV antibody responses in both the gut and circulation. For the sIgA levels of the I.G. immunized mice, a significant increase in antibody production was detected in the feces of the immunized mice with rNC8-LP3065-A104R, rNC8-LP3065-K78R, rNC8-LP3065-E120R, and rNC8-LP3065-E183L. Meanwhile, rNC8-LP3065-D117L and rNC8-LP3065-H171R remained insignificant. Conversely, sera from the constructs rNC8-LP3065-D117L and rNC8-LP3065-H171R were significant. In the BALF of the I.G. group, rNC8-LP3065-K78R, rNC8-LP3065-A104R, and rNC8-LP3065-E120R induced significant antibody levels, whereas rNC8-LP3065-183L and rNC8-LP3065-H171R elicited low antibody levels, and rNC8-LP3065-D117L showed insignificant results (Figure 4 and Figure 5).

Moreover, the mice immunized via the I.N. route with rNC8-LP3065-ASFV-mix exhibited significantly higher fecal sIgA levels than rNC8-pMG36e controls, as demonstrated by ELISA. The sIgA from feces showed that the mice immunized via the I.N. route with rNC8-LP3065-K78R and rNC8-LP3065-A104R produced higher levels of sIgA antibody levels in feces, while rNC8-LP3065-D117L generated low levels of sIgA antibodies. The immunizations with rNC8-LP3065-H171R, rNC8-LP3065-E120R, and rNC8-LP3065-E183L remained insignificant compared with the control. In the BALF of the I.N. immunized mice, there was a significant rise in sIgA levels in the mice immunized with the rNC8-LP3065-E120R, rNC8-LP3065-K78R, and rNC8-LP3065-A104R, while low levels of antibodies were detected in rNC8-LP3065-E183L and rNC8-LP3065-D117L. The immunization with construct rNC8-LP3065-H171R was insignificant, which suggests varying efficacy depending on the route of administration and the specific strains used (Figure 5).

### 3.4. rNC8-LP3065-ASFV-mix Induces T Cell Proliferation in Mice

To further assess the ability of rNC8-LP3065-ASFV-mix to induce cellular immune responses, flow cytometry was employed to evaluate T cell proliferation induced by rNC8-LP3065-ASFV-mix. The mice were immunized I.G. and I.N. with rNC8-LP3065-ASFV-mix, rNC8-pMG36e (negative control), or PBS (mock control). The spleens were collected following the euthanasia of the immunized mice.

CD3, CD4, and CD8 are surface markers on T cells that distinguish the cytotoxic CD8^+^ T cells and CD4^+^ helper T cells. These T cells also promote natural killer (NK) cell activity, antigen presentation, and enhance macrophage lysosomal activity. Flow cytometry analysis revealed a significant increase in CD3^+^CD4^+^ T cells in the spleens of the I.G. immunized mice with rNC8-LP3065-ASFV-mix, while CD3^+^CD8^+^ T cells were insignificant compared with the rNC8-pMG36e control, and in the I.N. immunized mice, both CD3^+^CD4^+^ and CD3^+^CD8^+^ T cells were insignificant compared with the control group. The I.G. immunized group induced more T cell proliferation compared with the I.N. administered group and PBS control group, indicating a significant cellular response in the I.G. immunized mice (Figure 6 and Appendix A).

### 3.5. I.G. and I.N. Immunization of rNC8-LP3065-ASFV-mix Increased Th1 Cytokine Levels and Partially Enhanced Th2 Cytokine Levels

To assess the immune responses induced by rNC8-LP3065-ASFV-mix in mice, serum levels of cytokines IL-2, IL-4, IL-10, and IFN-*γ* were quantified using ELISA. IFN-*γ* and IL-2, as pivotal Th1-associated cytokines, play crucial roles in the activation of cytotoxic T lymphocytes (CTLs), macrophages, and natural killer (NK) cells, while also promoting CD8^+^ T cell responses and enhancing innate immune functions. The results demonstrated distinct cytokine profiles among the immunization groups. Mice receiving rNC8-LP3065-ASFV-mix via either I.G. or I.N. routes showed significantly different levels of IFN-*γ* and IL-2 compared to both PBS and rNC8-pMG36e groups (Figure 7).

IL-4 and IL-10, as representatives of Th2 cytokines, promote B cell proliferation, antibody production, and plasma cell differentiation, while modulating inflammatory responses. In mice immunized with rNC8-LP3065-ASFV-mix, serum IL-4 levels were significantly elevated compared to those in the PBS and rNC8-pMG36e groups after I.G. and I.N. administration. In contrast, IL-10 levels did not differ significantly among the experimental groups, indicating that this cytokine response was not influenced by the immunization (Figure 7).

## 4. Discussion

Although swine production has improved significantly, infectious diseases remain a major barrier to industry sustainability, with ASFV posing the greatest threat. This highly virulent pathogen, for which no effective vaccines exist, causes severe hemorrhagic fever in domestic pigs and wild boars, leading to high mortality and major losses in the food chain. Its rapid spread across Africa, Europe, and Asia has disrupted trade and food security issues, underscoring the urgent need for effective vaccine development [19,20]. Adaptive immune responses, particularly antibody-mediated immunity, are essential for pathogen protection, emphasizing the importance of developing novel ASFV-targeted immunization [17,21]. Systemic and mucosal immunity are the primary defense mechanisms against invading infections in the body [22,23]. In animals, mucosal immunization is notable for its unique antigen-presenting ability [14,24]. The lumen of the intestine and the respiratory tract is a major carrier of immune cells in animals and is closely associated with the microenvironment. Maintaining a harmonious balance within the mucosal immune systems of the gut and nasal mucosa are of the utmost importance in protecting against pathogens. The mucosal immune system is uniquely able to generate effective protective immunity while maintaining immune homeostasis to prevent damaging inflammatory responses [25,26]. This is accomplished through the production of sIgA, the dominant antibody class in mucosal secretions. sIgA serves as a first line of defense by binding to pathogens and preventing their attachment to mucosal surfaces, thereby neutralizing them.

The cell wall anchoring mechanism enables the presentation of exogenous proteins on the surfaces of LAB, thereby improving immune recognition [27,28]. These surface proteins are covalently attached to the peptidoglycan layer via an amide bond formed between their C-terminal carboxyl group and the amino group of the pentaglycine cross-bridge [29]. Researchers reported the in vitro expression of ASFV proteins p30, p54, and p72 using the *Lactococcus lactis* system, constructing six recombinant strains expressing p30-LTB, p54-LTB, or p72-LTB fusion proteins. Immunization of rabbits with these strains led to significant increases in serum IgG, mucosal sIgA, cytokines IL-4 and IFN-*γ*, and splenocyte viability, enhancing mucosal, humoral, and Th2 immune responses. These results align with our findings [7]. Another study used the *L. lactis* surface display system to express rotavirus VP6 as a cell-wall-anchored fusion protein. Mice immunized subcutaneously with 10^9^ CFU/mL of *L. lactis* NZ9000/pCWA:VP6 developed rotavirus-specific antibodies by day 28, while control mice showed no antibody response [30]. In other studies, researchers successfully anchored the Venus antigen to the surface of *Limosilactobacillus fermentum* using a LysM-based anchoring domain. BALB/c mice immunized I.N. with the Venus-Acglu-BLPs027 complex exhibited significant increases in antigen-specific serum IgG, IgG1, IgG2a, and IgA levels in BALF. Additionally, spleen cells from immunized mice showed elevated production of TNF-*α*, IFN-*γ*, and IL-4, demonstrating the induction of both humoral and cellular immune responses [31]. Another study showed that intranasal co-administration of recombinant *L. lactis* and *L. plantarum* expressing HPV-16 E7 antigen induced IL-12 secretion and therapeutic effects against HPV-16 tumors in mice. Comparing I.G. and I.N. routes, the highest systemic responses were achieved via intranasal delivery, contrasting with our findings [32].

In this study, six ASFV antigens (E120R, E183L, D117L, H171R, K78R, and A104R) were selected based on their reactivity with convalescent animal sera and were successfully ligated into the pMG36e-LP3065 vector for surface display on NC8 [33,34]. To prevent hypersensitivity from live bacteria, the mixture was heat-inactivated by boiling in distilled water for 15 min, effectively eliminating viable cells while retaining essential antigenic structures. The cooled preparation was then administered via the I.V. route. No signs of hypersensitivity, such as local inflammation or systemic reactions, were observed during the immunization schedule, confirming the safety and effectiveness of the inactivation method. All recombinant *Lactobacillus* strains rNC8-LP3065-E120R/E183L/D117L/H171R/K78R/A104R proved to be effective and safe in all routes of administration. All the immunized mice remained in good health, and none of them showed clinical signs.

In the I.V. immunized mice, the ELISA results showed that the levels of the specific IgG antibodies against various proteins vary significantly due to the immune responses elicited by the specific antigens. The mice immunized with the recombinant strains rNC8-LP3065-K78R and rNC8-LP3065-A104R via the I.V. route showed a significant increase in specific IgG antibody titers and produced a potent immune response, while the recombinant strains rNC8-LP3065-E183L and rNC8-LP3065-H171R induced low antibody titers. However, amidst this variability, certain antigens exhibit remarkable efficacy in inducing IgG production, i.e., K78R and A104R. Moreover, rNC8-LP3065-K78R and rNC8-LP3065-A104R significantly increased IgG production by approximately three-fold in the mice, indicating that infection protection may be related to the production of specific antibodies against antigens compared with the I.G. and I.N. administered mice. The I.G. administration induced significant levels of sIgA compared with I.N. delivery, likely due to the complex immune architecture of the intestinal mucosa. Enhanced mucosal immunity arises from the gut-associated lymphoid tissue (GALT), a specialized component of the mucosa-associated lymphoid tissue (MALT) system, which possesses a highly organized immunological structure. This sophisticated arrangement underlies the superior sIgA response observed following I.G. immunization, in contrast to the less developed and more compartmentalized MALT associated with the I.N. route. Evolutionary continuous antigen exposure in the intestinal mucosa has promoted the development of this robust secretory immune system, making it particularly effective at generating mucosal antibody responses [35,36]. Secretory IgA (sIgA), structurally resistant to mucosal degradation, is produced via tightly regulated T-cell-independent and T-cell-dependent pathways to defend against microbes. Since most infections start at mucosal surfaces, vaccines should induce pathogen-specific sIgA alongside systemic immunity. The effectiveness of sIgA responses from I.G. or I.N. immunization depends on delivery systems that target immune cells and specific anatomical sites [7,32].

The safety characteristics of this recombinant vaccine system have attracted substantial attention. ELISA analyses have verified that the surface-expressed ASFV antigens effectively protected mice [37]. These encouraging results indicate that the platform could evoke comprehensive immune protection while maintaining an outstanding safety record [24,38].

Bacterial expression facilitates antigen production but lacks glycosylation, which must be considered when interpreting immunogenicity. For ASFV, comparing responses to glycosylated versus non-glycosylated antigens will clarify the role of glycosylation in protection. Hybrid platforms incorporating eukaryotic glycosylation could retain the benefits of the LP3065 anchoring system. Future studies should assess the rNC8 system in pigs and investigate glycosylation’s impact on ASFV proteins to identify key protective antigens.

## 5. Conclusions

Taken together, the surface display of the ASFV antigens could increase the adaptive immune responses in a mouse model by increasing the specific antibodies and CD3^+^CD4^+^ and CD3^+^CD8^+^ T cell responses. Our findings imply that rNC8 exhibited a significant rise in serum IgG antibodies in the immunized mice. The rNC8 also increased the CD3^+^CD4^+^ T cell activation in the I.N. group and CD3^+^CD8^+^ T cells in both the I.N. and I.G. groups.

## Figures and Tables

**Figure 1 vaccines-13-00897-f001:**
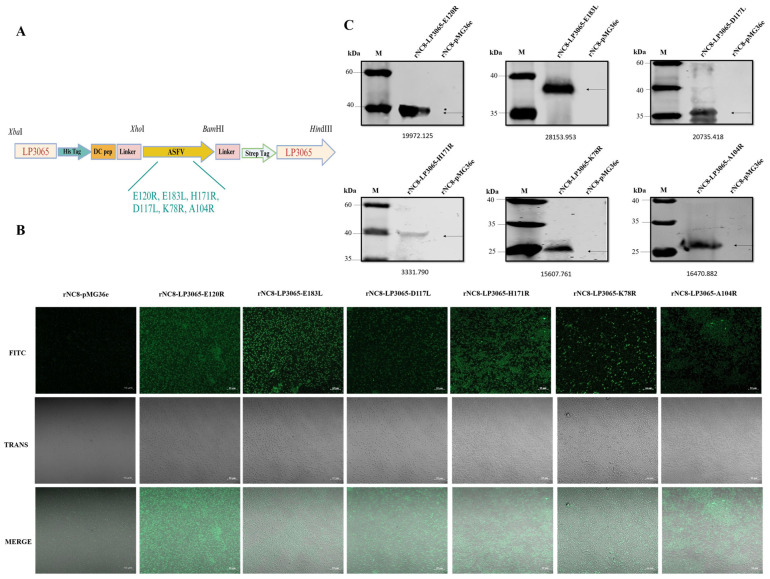
Construction and characterization of recombinant rNC8-LP3065 expressing the ASFV antigen mix. (**A**) Schematic recombinant plasmid construction pMG36e-LP3065-E120R/E183L/D117L/H171R/K78R/A104R. (**B**) Identification of the recombinant rNC8-LP3065-ASFV-mix by IFA. Scale bar = 10 µm. (**C**) Western blotting analysis of antigen expression in the rNC8-LP3065-ASFV-mix.

**Figure 2 vaccines-13-00897-f002:**
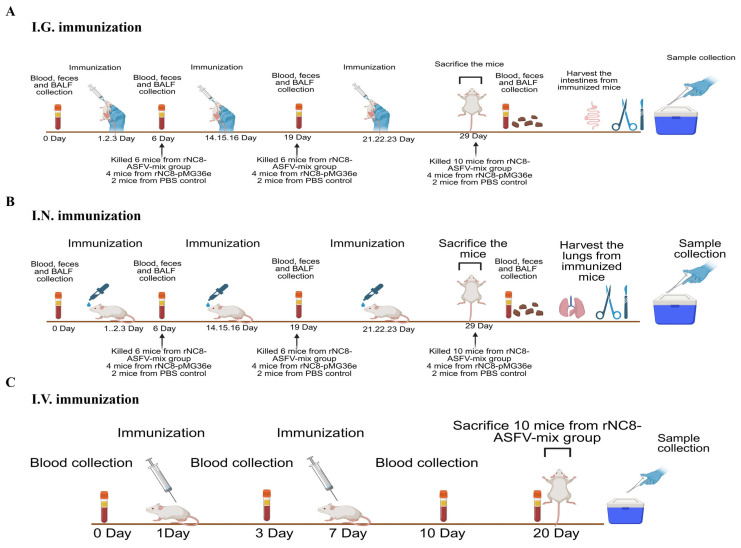
Schematic overview of rNC8-LP3065-ASFV-mix immunization in mice. Diagrams illustrating I.G. (**A**), I.N. (**B**), and I.V. (**C**) immunization routes are presented. The mice were immunized with a heat-inactivated bacterial mix of 200 µL (10^9^ CFU/strain/mouse) via the I.V. and I.G. routes, or with 20 µL (10^9^ CFU/strain/mouse) via the I.N. route, at the designated time points. Sera samples were collected on days0, day 6, day 19, and day 29 dpi for the I.G. and I.N. groups and on day 0, day 3, day 10, and day 20 dpi for the I.V. group.

**Figure 3 vaccines-13-00897-f003:**
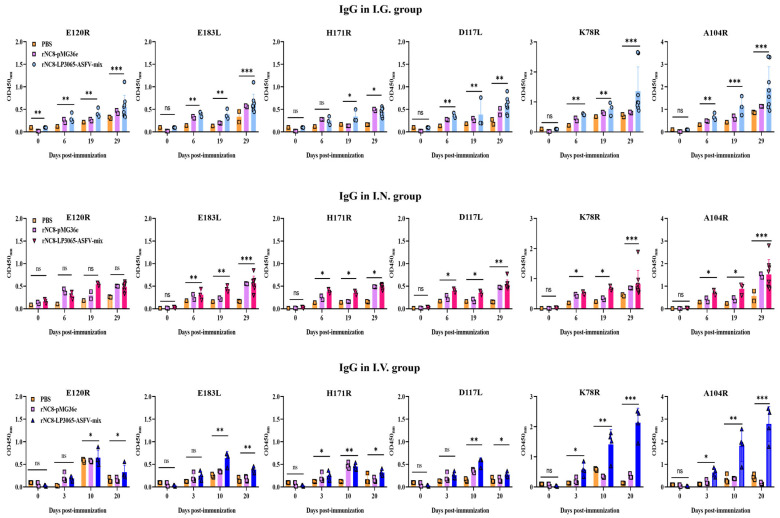
ELISA-based detection of ASFV-specific IgG antibodies in mice immunized with rNC8-LP3065-ASFV-mix. Bars represent means ± SDs of the independent experiments; ns: not significant, *p* ≥ 0.05, * *p* < 0.05, ** *p* < 0.01, *** *p* < 0.001.

**Figure 4 vaccines-13-00897-f004:**
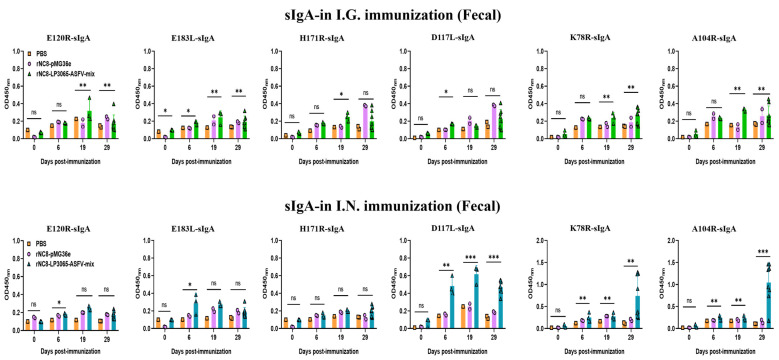
Detection of sIgA antibodies in fecal samples from rNC8-LP3065-ASFV-mix-immunized mice by ELISA. Bars represent SDs of three independent experiments; ns: not significant *p* ≥ 0.05, * *p* < 0.05, ** *p* < 0.01, *** *p* < 0.001.

**Figure 5 vaccines-13-00897-f005:**
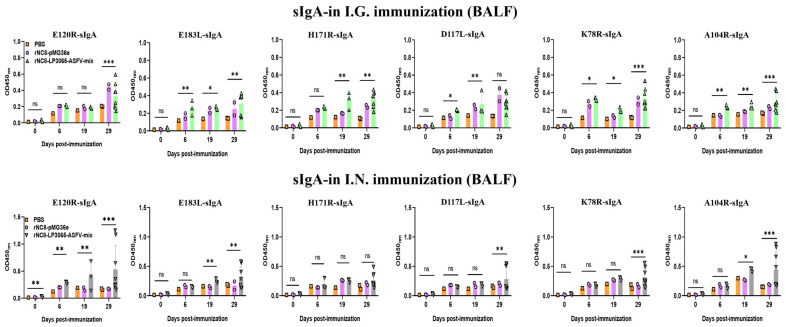
Specific sIgA antibodies in BALF of rNC8-LP3065-ASFV-mix immunized mice by ELISA. Bars represent SDs of three independent experiments; ns: not significant *p* ≥ 0.05, * *p* < 0.05, ** *p* < 0.01, *** *p* < 0.001.

**Figure 6 vaccines-13-00897-f006:**
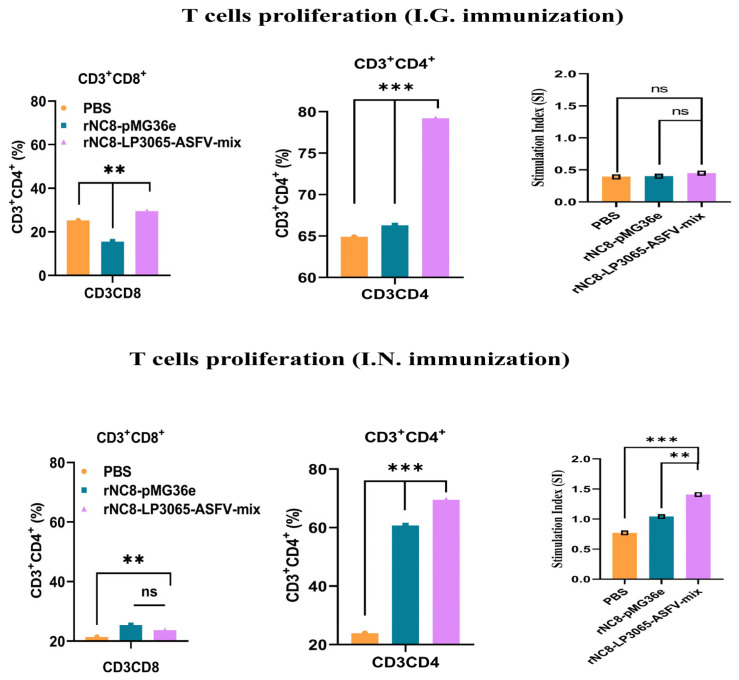
Assessment of T cell proliferation and subset distribution in mice immunized with rNC8-LP3065-ASFV-mix. Proliferative activity was measured using the CCK-8 assay. The expression of CD3^+^, CD4^+^, and CD8^+^ molecules on T cells was analyzed by flow cytometry. Bars represent SDs of three independent experiments; ns: not significant *p* ≥ 0.05, ** *p* < 0.01, *** *p* < 0.001.

**Figure 7 vaccines-13-00897-f007:**
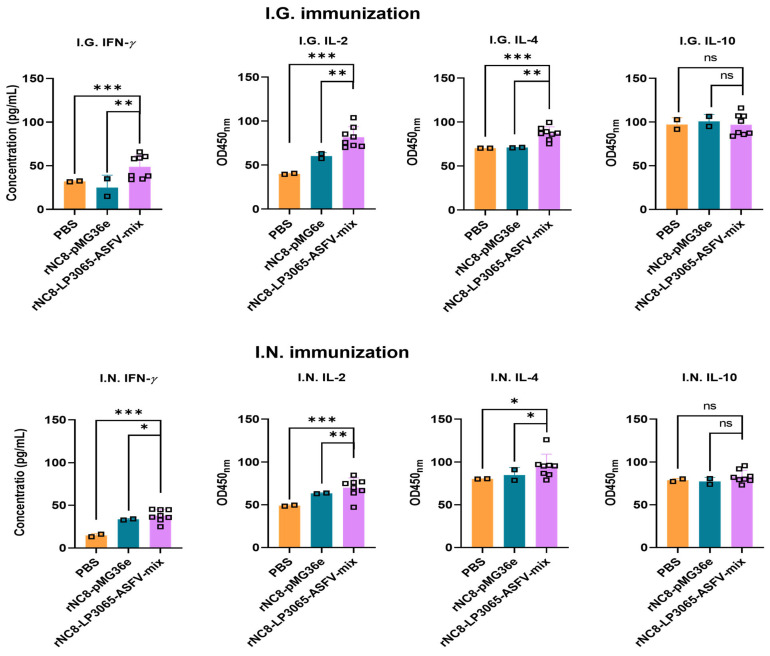
The levels of Th1 (IFN-*γ* and IL-2) and Th2 (IL-4 and IL-10) cytokines in the sera of the I.G. or I.N. immunized mice; bars represent SDs of three independent experiments; ns: not significant *p* ≥ 0.05, * *p* < 0.05, ** *p* < 0.01, *** *p* < 0.001.

**Table 1 vaccines-13-00897-t001:** Sequences of the primers used in this study.

Primers	Sequence ^a^ (5′-3′)
E120R-F	ACCGCTCGAGATGGCTGATTTTAATTCACCAATTC
E120R-R	CGGGATCCGCCGCCTTTAGATTTATGAGATG
E183L-F	ACCGCTCGAGATGGATTCAGAATTTTTCCAACCA
E183L-R	CGGGATCCGCCGCCAAGAGAATTTTCTAAATC
D117L-F	ACCGCTCGAGATGGATACAGAAACTTCACCTCTTCTT
D117L-R	CGGGATCCGCCGCCTGAATGTGCAAGTTCAG
H171R-F	ACCGCTCGAGATGGTTGTTTATGATCTTCTTGTTTCA
H171R-R	CGGGATCCGCCGCCATTTTTAATAGAAAACA
K78R-F	ACCGCTCGAGATGCCTACAAAAGCTGGCACAAAAAGTACCGC
K78R-R	CGGGATCCTTTTGACCGTTTAATTTTTTT
A104R-F	ACCGCTCGAGATGTCGACAAAAAAAAAGCCCACAATTACCAAGCAAGA
A104R-R	CGGGATCCGTTTAACATATCATGGACAGGT

^a^ The underlined nucleotides are the positions that anneal to the vector.

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
