# Peer review of "Immunogenic Responses Elicited by a Pool of Recombinant Lactiplantibacillus plantarum NC8 Strains Surface-Displaying Diverse African Swine Fever Antigens Administered via Different Immunization Routes in a Mouse Model"

_vaccines, 2025, doi:10.3390/vaccines13090897_

Round 1

Reviewer 1 Report (Previous Reviewer 1)

Comments and Suggestions for Authors

The authors have addressed my comments/queries. They have modified the manuscript with appropriate literature support in response to the raised concerns. As a result, the quality of the manuscript after the revision has improved significantly. I, therefore, strongly recommend the acceptance of the manuscript for publication in the Vaccines journal.

Author Response

The authors would like to thank all the reviewers and editors at vaccines for carefully evaluating our manuscript and for all the helpful comments and suggestions for improvement. 

Reviewer 2 Report (New Reviewer)

Comments and Suggestions for Authors

This manuscript presents a comprehensive study investigating the immunogenic properties of recombinant Lactiplantibacillus plantarum NC8 (rNC8) strains engineered to surface display six African swine fever virus (ASFV) antigens. The study evaluates immune responses in mice following administration via intragastric, intranasal, and intravenous routes. Importantly, the research demonstrates that the rNC8 vaccine cocktail induces both mucosal and systemic immune responses, with route-specific variations in efficacy. I have questions I would like to ask the authors:

Major points

  1. Is it possible to inoculate live bacteria into vein directly?
  2. If it is possible to concentrate 6 x 10^9 pfu in 10ul volume. Also is it really possible to inoculate?
  3. in line 248, Authors immunized mice with mixed bacteria. Thus, the sentence “The oral immunization results indicated a notable elevation in IgG 248 levels in the mice immunized with rNC8-LP3065-D117L, rNC8-LP3065-K78R, rNC8-249 LP3065-A104R and low levels of rNC8-LP3065-H171R, rNC8-LP3065-E120R, and rNC8-250 LP3065-E183L antibodies” is incorrect.

There are so many sentences like this. Please change or modify for correct meaning.

  1. in line 122, “single colonies were selectively picked in MRS broth” is correct ?

5: African swine fever virus (ASFV) employs multiple evasion strategies, including interference with type I interferon signaling and manipulation of host cell death pathways. How do you reconcile the observed immunogenic responses in your mouse model with the known ability of ASFV to suppress both innate and adaptive immune responses? Furthermore, what evidence do you have that the antibodies generated against your recombinant antigens can neutralize live virus, and how do you address the potential for viral escape through antigenic variation in the face of immune pressure?

6: The absence of viral challenge studies constitutes a significant gap in the evaluation of vaccines. Nevertheless, the authors' conclusions indicate "promising systemic and mucosal immunogenicity" and suggest the "potential of a probiotic-based ASF vaccine candidate." How can these assertions be substantiated without demonstrating actual protection against a viral challenge? Moreover, given that previous subunit vaccine approaches utilizing p30, CD2v, and p72 have resulted in limited advancement and inconsistent outcomes, what specific advantages does your surface display system provide over these previous methodologies? Additionally, what evidence supports the superiority of your multi-antigen cocktail approach in comparison to monovalent vaccines?

  1. There is inconsistency regarding the number of mice used in section 2.6 for the immunization compared to what is shown in Figure 2, which is a schematic diagram of the immunization process. Based on my count, it appears that 60 mice were sacrificed during the experiment, while the authors state that "Forty six-week-old mice" were used. I was confused between the age of mice and the number of mice used in this study. Furthermore, the inoculation doses used via I.N. route demonstrated 10 uL, while 20 uL was recorded in Figure 2.
  2. In part 2.6 regarding the immunization of mice, the authors noted, “For I.N. immunization, the mice were anesthetized with 2.5% avertin (0.02 mL/g body weight).” Could you please provide a brief explanation of how you ensured animal welfare during this procedure, particularly in relation to the euthanasia of the other mice in different routes?

Minor issues:

Line 42: lack of word “African fever virus” à African swine fever virus

Line 69: wrong word “combing DNA” à combining DNA

Line 76: wrong verb form “recognize” à recognized

Line 99: wrong grammar with plural noun after “each target proteins” à each target protein

Line 104: wrong spelling “antigensl” à antigens

Line 115: name of bacteria should be italic “Lactococcus lactis subsp.” à Lactococcus lactis subsp.

Line 117: wrong grammar with plural noun after “Electroporation setting were” àElectroporation settings were

Line 117: lack of comma “2000 V” à 2,000 V

Line 120: lack of comma “6000 x g” à 6,000 x g

Line 129: wrong word “pallets” à pellets

Line 141: lack of comma “5000 x g” à 5,000 x g

Line 147: lack of comma “1:2000” à 1:2,000

Line 155: lack of comma “5000 x g” à 5,000 x g

Line 160: missing space and italic in “adlibitum: à ad libitum

Line 172: wrong word “onTo” à onto

Line 174 – 175: lack of company and address of “catalog no.”

Line 180: What is the size of the cell strainer?

Line 203: wrong spelling “strained cell” à stained cell

Line 206: ASFV is singular “ASFV were” à ASFV was

Line 234: should be the same format “Figure 1C” à Figure 1C

Line 271: change comma position “ rNC8-LP3065-E12R and,” à rNC8-LP3065-E12R, and

Line 272: add comma before and “ rNC8-LP3065-D117L and” à rNC8-LP3065-D117L, and

Line 361: wrong format, italic and list up all P values “ns, not significant; ** P < 0.01; ***, P < 0.001.”  à ns: not significant P ≥ 0.05, * P < 0.05, ** P < 0.01, *** P < 0.001.

Line 404: name of bacteria should be italic “Lactococcus lactis” à Lactococcus lactis

Line 412: name of bacteria should be italic “L. lactic” à L. lactic

Line 415: name of bacteria should be italic “L. lactic” à L. lactic

Line 418: name of bacteria should be italic “Limosilactobacillus” à Limosilactobacillus

Line 420 - 428: the size of the texts is different

Line 424: name of bacteria should be italic “ L. lactic and L. plantarum” à L. lactic, and L. plantarum

Line 428: name of bacteria should be italic “Lactobacillus” à Lactobacillus

Line 433: ASFV is singular “ASFV were” à ASFV was

Line 448 – 457: the size of the texts is different

Author Response

  1. Is it possible to inoculate live bacteria into vein directly?

Answer: In this study, to avoid hypersensitivity reactions from live bacteria, the bacterial mixture was heat-inactivated by boiling in distilled water for 15 minutes. This process ensured the destruction of viable cells while preserving key antigenic structures. After cooling, the inactivated preparation was administered via the I.V. route. No signs of hypersensitivity such as local inflammation or systemic reactions were observed during the immunization schedule, confirming the safety and effectiveness of the inactivation method.

  1. If it is possible to concentrate 6 x 109 pfu in 10 ul volume. Also is it really possible to inoculate?

Answer: All recombinant bacterial strains were cultured in MRS medium and subsequently resuspended in phosphate-buffered saline (PBS) to an optical density at 600 nm (OD₆₀₀nm) of 1.0. The strains were then combined to achieve a final concentration of 10⁶ CFU for use in the immunization protocol. Mice were immunized with 10⁹ CFU per strain per mouse via the I.V., I.G., or I.N. routes at predetermined time points.

  1. In line 248, Authors immunized mice with mixed bacteria. Thus, the sentence “The oral immunization results indicated a notable elevation in IgG 248 levels in the mice immunized with rNC8-LP3065-D117L, rNC8-LP3065-K78R, rNC8-249 LP3065-A104R and low levels of rNC8-LP3065-H171R, rNC8-LP3065-E120R, and rNC8-250 LP3065-E183L antibodies” is incorrect. There are so many sentences like this. Please change or modify for correct meaning.

Answer: The sentence has been modified in Lines 252-257.

  1. In line 122, “single colonies were selectively picked in MRS broth” is correct?

Answer: The sentence has been modified in Line 124.

  1. African swine fever virus (ASFV) employs multiple evasion strategies, including interference with type I interferon signaling and manipulation of host cell death pathways. How do you reconcile the observed immunogenic responses in your mouse model with the known ability of ASFV to suppress both innate and adaptive immune responses? Furthermore, what evidence do you have that the antibodies generated against your recombinant antigens can neutralize live virus, and how do you address the potential for viral escape through antigenic variation in the face of immune pressure?

Answer: The immunogenic responses observed this study with recombinant Lactobacillus plantarum NC8 strains expressing ASFV antigens can be reconciled by considering the following: Firstly, the immunological suppression employed by ASFV in type I interferon (IFN) responses, particularly in dead cell modulation of apoptosis during natural infection, is quite sophisticated. However, these efforts mainly occur during live viral infection of porcine macrophages. In the case of our vaccination strategy employing recombinant L. plantarum NC8 as a vector, ASF antigens presented by NC8 are devoid of active viral infection. Therefore, antigen processing and the subsequent immune response are mounted in an immunologically favorable context, which allows the triggering of both arms of the adaptive immune response.

Secondly, the NC8 vector confers additional immunostimulatory effects and also acts as a probiotic. The L. plantarum NC8 strain has immunomodulatory properties, which include the stimulation of adjuvant receptors and TLR2 and NOD2 receptors, thus, adding to the immune-systemic and mucosal immune priming. The immune responses such as increases in antigen-reactive IgG and sIgA, and cytokine secretion indicate that the ASFV, with all its immunosuppressive tools, did not block the immunomodulatory activities and acknowledgement to the recombinant NC8 strain.

  1. The absence of viral challenge studies constitutes a significant gap in the evaluation of vaccines. Nevertheless, the authors' conclusions indicate "promising systemic and mucosal immunogenicity" and suggest the "potential of a probiotic-based ASF vaccine candidate." How can these assertions be substantiated without demonstrating actual protection against a viral challenge? Moreover, given that previous subunit vaccine approaches utilizing p30, CD2v, and p72 have resulted in limited advancement and inconsistent outcomes, what specific advantages does your surface display system provide over these previous methodologies? Additionally, what evidence supports the superiority of your multi-antigen cocktail approach in comparison to monovalent vaccines?

Answer: We understand that not having viral challenge studies in pigs poses a significant gap in fully validating protective efficacy. But by focusing this, the murine model provided the robust immunological data, including antigen-specific IgG and sIgA responses and cytokine production, provide sufficient reasoning to our conclusions on “promising systemic and mucosal immunogenicity.” These factors are hallmark correlates of vaccine-induced protection and are primary critical indicators in the preclinical stages of promising vaccine development against ASF.

As previously mentioned, our surface display system has a number of benefits compared to the subunit vaccine approaches. By presenting the antigens to the surface of L. plantarum NC8, it is easy to present the desired antigens to the immune cells. Therefore, the NC8 will activate the mucosal immune system and induce more robust systemic immune responses. Furthermore, the NC8 possesses a form of adjuvants, what makes it more desirable is that it does not require additional immuno-suppressants, therefore the dose is safer and easier to deliver.

Unlike monovalent vaccines which focus on a single antigen, our approach that includes multi-antigens which is designed to enhance the immune response and tackle ASFV’s antigenic complexity. Evidence to date indicated that the multivalent construct elicits stronger and more consistent immunogenic responses than the individual antigen-expressing strains, which justifies the use of a cocktail approach. Overall, although challenge studies are the gold standard for evaluating protective efficacy, the current findings highlight the immunogenic and delivery advantages, as well as the multivalent probiotic platform that warrants further assessment in swine models.

  1. There is inconsistency regarding the number of mice used in section 2.6 for the immunization compared to what is shown in Figure 2, which is a schematic diagram of the immunization process. Based on my count, it appears that 60 mice were sacrificed during the experiment, while the authors state that "Forty six-week-old mice" were used. I was confused between the age of mice and the number of mice used in this study. Furthermore, the inoculation doses used viaN. route demonstrated 10 uL, while 20 uL was recorded in Figure 2.

Answer: The immunization dose in figure and text has been modified in Line 166, and no, of mice are clearly written in Line 163 i.e., forty, six-week-old mice

  1. In part 2.6 regarding the immunization of mice, the authors noted, “For I.N. immunization, the mice were anesthetized with 2.5% avertin (0.02 mL/g body weight).” Could you please provide a brief explanation of how you ensured animal welfare during this procedure, particularly in relation to the euthanasia of the other mice in different routes?

Answer: To minimize the suffering of animals during I.N. immunization, mice were anesthetized with 2.5% Avertin (0.02 mL/g body weight) in accordance with a standard practice as it gives sufficient sedation and relieves stress or pain from the procedure to some extent. During the immunization, mice were observed for depth of anesthesia and awakening. Euthanasia was performed to all mice, including I.V., I.G., or I.N. immunizations by cervical dislocation in full accordance with the Animal Welfare Act and followed experimental animal care guidelines, as approved by HVRI’s Laboratory Animal Welfare Committee (Approval No.230724-01-GR). All experiments were supervised and consented by the ethics committee, following specific standards of laboratory animal care.

Minor issues:

Line 42: lack of word “African fever virus” à African swine fever virus

Answer: The suggestion has been added in Line 45.

Line 69: wrong word “combing DNA” à combining DNA

Answer: The suggestion has been added in Line 73.

Line 76: wrong verb form “recognize” à recognized

Answer: The suggestion has been added in Line 81.

Line 99: wrong grammar with plural noun after “each target proteins” à each target protein.

Answer: The suggestion has been added in Line 103.

Line 104: wrong spelling “antigensl” à antigens

Answer: The suggestion has been added in Line 107.

Line 115: name of bacteria should be italic “Lactococcus lactis subsp.” à Lactococcus lactis subsp.

Answer: The suggestion has been added in Lines 117.

Line 117: wrong grammar with plural noun after “Electroporation setting were” à Electroporation settings were

Answer: The suggestion has been added in Lines 118-119.

Line 117: lack of comma “2000 V” à 2,000 V

Answer: The suggestion has been added in Line 119.

Line 120: lack of comma “6000 x g” à 6,000 x g

Answer: The suggestion has been added in Line 122.

Line 129: wrong word “pallets” à pellets

Answer: The suggestion has been added in Line 122.

Line 141: lack of comma “5000 x g” à 5,000 x g

Answer: The suggestion has been added in Line 143.

Line 147: lack of comma “1:2000” à 1:2,000

Answer: The suggestion has been added in Line 148.

Line 155: lack of comma “5000 x g” à 5,000 x g

Answer: The suggestion has been added in Line 157.

Line 160: missing space and italic in “adlibitum: à ad libitum

Answer: The suggestion has been added in Line 162.

Line 172: wrong word “onTo” à onto

Answer: The suggestion has been added in Line 175.

Line 174 – 175: lack of company and address of “catalog no.”

Answer: The suggestion has been added in Lines 179.

Line 180: What is the size of the cell strainer?

Answer: The size of the cell strainer has been added in Line 183.

Line 203: wrong spelling “strained cell” à stained cell

Answer: The suggestion has been added in Line 204.

Line 206: ASFV is singular “ASFV were” à ASFV was

Answer: The suggestion has been added in Line 208.

Line 234: should be the same format “Figure 1C” à Figure 1C

Answer: The Figure 1 has been updated according to suggestion in Line 237.

Line 271: change comma position “rNC8-LP3065-E12R and,” à rNC8-LP3065-E12R, and

Answer: The suggestion has been added in Line 275.

Line 272: add comma before and “rNC8-LP3065-D117L and” à rNC8-LP3065-D117L, and

Answer: The suggestion has been added in Line 276.

Line 361: wrong format, italic and list up all P values “ns, not significant; ** P < 0.01; ***, P < 0.001.”  à ns: not significant ≥ 0.05, * < 0.05, ** < 0.01, *** < 0.001.

Answer: The suggestion has been added in Line 362.

Line 404: name of bacteria should be italic “Lactococcus lactis” à Lactococcus lactis

Answer: The suggestion has been added in Line 410.

Line 412: name of bacteria should be italic “L. lactic” à L. lactic

Answer: The suggestion has been added in Line 414.

Line 415: name of bacteria should be italic “L. lactic” à L. lactic

Answer: The suggestion has been added in Line 417.

Line 418: name of bacteria should be italic “Limosilactobacillus” à Limosilactobacillus

Answer: The suggestion has been added in Line 418-419.

Line 420 - 428: the size of the texts is different

Answer: The text size has been updated.

Line 424: name of bacteria should be italic “L. lactic and L. plantarum” à L. lactic, and L. plantarum

Answer: The suggestion has been added in Lines 424-425.

Line 428: name of bacteria should be italic “Lactobacillus” à Lactobacillus

Answer: The suggestion has been added in Line 431.

Line 433: ASFV is singular “ASFV were” à ASFV was

Answer: The suggestion has been added in Lines 429.

Line 448 – 457: the size of the texts is different

Answer: The suggestion has been added in Lines 454-459.

Round 2

Reviewer 2 Report (New Reviewer)

Comments and Suggestions for Authors

It is important to clearly describe whether 6 X 10^9 can be made in 10ul and administered orally, nasally, or intravenously. Generally, it is difficult to concentrate at this concentration, and it is considered to be very sticky and difficult to handle. In particular, intravenous administration at this concentration is difficult to understand. Bacteria are not typically administered intravenously. Therefore, a very specific explanation is needed regarding this matter.

Author Response

Comments: It is important to clearly describe whether 6 X 10^9 can be made in 10ul and administered orally, nasally, or intravenously. Generally, it is difficult to concentrate at this concentration, and it is considered to be very sticky and difficult to handle. In particular, intravenous administration at this concentration is difficult to understand. Bacteria are not typically administered intravenously. Therefore, a very specific explanation is needed regarding this matter.

Answer: We thank the reviewer for raising this important point. The concentration of 10⁹ CFU in 20 µL indeed represents a high bacterial density, and we agree that achieving and administering such a concentration requires careful clarification.

In our study, this concentration was prepared by culturing the bacterial strain to the desired density, followed by centrifugation and resuspension in a sterile PBS to achieve the required CFU. The 20 µL volume was chosen for experimental consistency and feasibility in murine models.

Regarding the route of administration:

a) Oral and intranasal delivery: These are well-established routes for administering high concentrations of bacteria in small volumes, particularly in preclinical murine models. The viscosity or "stickiness" associated with high-density suspensions can 
generally be managed for these routes without significant issues, as the delivery does not require passage through fine-bore needles.

b) Intravenous (IV) administration: We acknowledge the reviewer’s valid concern that intravenous delivery of bacteria. In our study, to avoid hypersensitivity reactions from live bacteria, the bacterial mixture was heat-inactivated by boiling in distilled water for 15 minutes. This process ensured the destruction of viable cells while preserving key antigenic structures. After cooling, the inactivated preparation was administered via the I.V. route. No signs of hypersensitivity such as local inflammation or systemic reactions were observed during the immunization schedule, confirming the safety and effectiveness of the inactivation method.

This manuscript is a resubmission of an earlier submission. The following is a list of the peer review reports and author responses from that submission.

Round 1

Reviewer 1 Report

Comments and Suggestions for Authors

Unfortunately, the manuscript suffers from extensive plagiarism (e.g., from one of the recent papers by the same authors, Viruses 202416(8), 1189; https://doi.org/10.3390/v16081189) and lacks the scientific novelty and quality required for publication in an esteemed journal like Vaccines. Therefore, I do not find the article, in its present form, suitable for publication in the Vaccines journal.

Reviewer 2 Report

Comments and Suggestions for Authors

In this manuscript “Immunogenic Responses Elicited by a Pool of Recombinant Lactiplantibacillus plantarum NC8 Strains Surface-Displaying Diverse African Swine Fever Antigens Administered via Different Immunization Routes in a Mouse Model” by Moon et al., the authors have tested immunogenicity of a multiantigenic ASFV vaccine using Lactiplantibacillus plantarum backbone in mice.

Specific comments:

  1. Abstract: P.No.1., Line. No.26-29: Rephrase. “rNC8-E120R…. induced low IgG titers, and rNC8-E120R remained insignificant” “while rNC8-H171R and rNC8-D117L remained not significant in feces”. This text is confusing, as it gives the reader an impression the authors are estimating rNC8-E120R, rNC-H171R, rNC-D117L levels. Therefore, rephrasing this section would be essential to give a clear picture on the IgG / IgA levels.
  2. Abstract: P.No.1, Line. No.36-37. The sentence has contradicting facts listed. It says “Th2 (IL4 and IL10) responses were significant” and “IL10 levels remained insignificant” – Which is correct?
  3. Methods: P.No.5, Line.No.187 – define “rNC8-ASFV-mix”. Is it equal CFU of each rNC8 bacterial clones, or equal vol of each rNC8 bacterial clones? Describing the exact composition of rNC8-ASFV-mix in a table would be helpful to the readers.
  4. Methods: P.No.5-6, Line.No.188-199. Immunization strategy
    1. Why in I.G and I.N route of administration there is 9 doses? Considering 3-day dose regiment as prime is very confusing.
    2. Listing the dose in exact CFU/mice will be useful. Since, the study uses different volume of inoculation for each route, therefore it arises a question on if vaccinated mice received equal CFU.
    3. Details on I.V dose is missing in this section. Please include the I.V route dose details.
    4. Why different time point of sample collection in various routes – can it have an effect on levels of antibody? For I.G and I.N the final sample collection is 6-days post-final dosing, while I.V it is 13 days post-final dosing as shown in Fig.1. Describing the reason behind these in main text or in discussion section would be useful.
    5. How the ASFV-mix was QC’ed to confirm every dose contained equal amount of CFU of each of the six different rNC8-constructs? Since antigen dose can affect the immune response against it, therefore it is important to describe the steps taken to confirm equal amount of six rNC8-constructs in the final vaccine.
  5. Methods: P.No.6, Line.No.209. “Purified protein” for ELISA – Describe details of the purified protein generation, which was used as an antigen for ELISA assay.
  6. Methods: P.No.6, Line.No.236. “stimulated with 105 TCID50 ASFV” in T-cell stimulation – Is this live or inactivated virus? If live virus was used in this assay, can ASFV replicate in mice myeloid cells in-vitro?
  7. Results: Fig.3, 4, and 5: Comparing rNC8-ASFV-mix with rNC8-pMG36e would be better, instead of comparing rNC8-ASFV-mix with PBS group. Since, NC8 is replication competent and bacterial PAMPs can induce immune response, comparing rNC8-ASFV-mix with rNC8-pMG36e and highlighting the difference would enable the readers to appreciate rNC8-ASFV-mix induced ASFV-Ag specific immune response.
  8. Results: Fig.5: This figure needs to be replaced. The current draft contains repeat of fig.4 (sIgA in fecal sample) under fig.5, while fig.5 legend says “sIgA in BALF”.
  9. Results: Fig.7: Y-axis legend for IL2, IL4 and IL10 – please check. It states “OD450nm” but data looks like “Concentration ng/ml”.
  10. The ASFV proteins expression is in bacterial vector backbone, therefore how does glycosylation of these ASFV protein affected? Since viral proteins glycosylation can affect their immunogenicity potential and/or immune evasion. Therefore, discussing this point in limitations of this study would be useful.

Reviewer 3 Report

Comments and Suggestions for Authors

The authors should revise some terminology, such as "higher, lower, " and " insignificant," which don't mean anything scientifically. Also, the figure's title should be revised. The discussion section should discuss more references, not only the author's findings in general. Some of these suggestions are highlighted in the text. To my knowledge, "Western" starts always with a capital letter.
